# Isolation of Fungal Strains from Municipal Wastewater for the Removal of Pharmaceutical Substances

**Brigita Dalecka [1,2,\*]**, **Caroline Oskarsson [1]**, **Talis Juhna [2]** and **Gunaratna Kuttava Rajarao [1]**

[1]   School of Engineering Science in Chemistry, Biotechnology and Health, Albanova University Centre, KTH Royal Institute of Technology, Roslagstullcbacken 21, 114 21 Stockholm, Sweden; carosk@kth.se (C.O.); gkr@kth.se (G.K.R.)

[2]   Water Research Laboratory, Riga Technical University, Kipsala 6A-263, LV-1048 Riga, Latvia; talis.juhna@rtu.lv

\*   Correspondence: brigita@kth.se; Tel.: +371-26553115

**Abstract:** Fungi have been shown to be promising candidates to be used in removal of pharmaceutical compounds during wastewater treatment processes. However, fungal growth, including removal efficiency, can be affected by several factors, such as temperature and the pH. The ability of fungal isolates to grow in the presence of carbamazepine, diclofenac, ibuprofen, and sulfamethoxazole was tested. Removal efficiency results indicated that a fungal isolate of *Aspergillus luchuensis* can completely (>99.9%) remove diclofenac from a synthetic wastewater media without a pH correction within 10 days of incubation. Furthermore, the results of the biosorption test for *A. luchuensis* indicate that this isolate uses the biosorption mechanism as a strategy to remove diclofenac. Finally, the results demonstrate that *A. luchuensis* can remove >98% of diclofenac in non-sterile wastewater without a pH correction immediately after biomass inoculation on biofilm carriers while *Trametes versicolor* requires an incubation period of at least 24 h to completely remove diclofenac. Therefore, this isolate is a promising candidate for use in removal of pharmaceutical compounds from wastewater with typical pH 7.8, minimizing a requirement of the pH correction.

**Keywords:** wastewater; fungi; *Trametes versicolor*; *Aspergillus luchuensis*; diclofenac; carbamazepine

---

## 1. Introduction

Conventional wastewater treatment technologies cannot always remove pharmaceutical substances efficiently [1]. In such cases, the pharmaceuticals are released into the environment where they negatively affect living organisms. [2,3]. Alternative technologies used in methods to remove pharmaceutical substances from wastewater such as advanced oxidation, UV disinfection, ozonation, and granular activated carbon progressing still cause concerns around by-product formation and the cost of energy and chemical consumption [4–6]. Therefore, research into biological treatments to remove pharmaceutical substances using fungi might be an attractive topic, with an aim to develop effective and environmentally friendly wastewater treatment technology.

White-rot fungi have shown to be good candidates to remove pharmaceutical substances from wastewater [7]. They are able to degrade a wide variety of pharmaceuticals due to their enzymatic processes. Furthermore, fungi can use a biosorption strategy to counteract the effects of pharmaceuticals because of their specific cell wall composition, formed from chitosan and chitin [8,9]. In addition, *Trametes versicolor* can metabolize and integrate some pharmaceuticals, such as benzophenone-3 and diclofenac, into amino acids of fungi [10,11]. However, the growth of specific fungal strains can be affected by several factors such as the pH, temperature, concentration of inhibitory substances,

and nutrients [12]. Therefore, fungi isolated from municipal wastewater treatment plants are more likely to be adapted to the environmental and operating conditions [13]. Thus, determination of the removal potential of residential wastewater fungi is an important task in developing industrial process applications in order to accomplish the long-term goal of pharmaceutical removal. Furthermore, when using low-cost materials such as fungi, advantages includes a low capital investment, relatively simple operations, low operating costs, and the lack of degradation by-products [14,15].

The main focus of this research was to investigate the isolation of fungal strains from municipal wastewater, and to test their ability to remove pharmaceutical substances. To achieve this, fungal isolates were isolated and cultivated on a synthetic wastewater media in the presence of selected pharmaceuticals. During this study, the effect of the pH on removal efficiency was studied. The most promising isolate was further identified and analyzed in non-sterile municipal wastewater. Finally, a biosorption experiment was conducted with the isolate, and enzyme activity was measured to better understand the removal mechanisms of pharmaceutical substances. All results of fungal isolate were compared to *T. versicolor* to evaluate the potential of an isolated fungal strain and the advantages of its application in wastewater treatment to remove pharmaceutical substances.

## 2. Materials and Methods

### 2.1. Fungi, Pharmaceuticals Substances and Wastewater

The white-rot fungus of *Trametes versicolor* DSM 6401 was obtained from the culture collection (Leibniz Institute DSMZ—German Collection of Microorganisms and Cell Cultures, Brunswick, Germany) and used as a control strain. The standard solutions at the concentration of 5 mg/L of carbamazepine (CAR), diclofenac (DIC), ibuprofen (IBU), and sulfamethoxazole (SUL) (Sigma-Aldrich, Taufkirchen, Germany) were prepared separately, according to the manufacturer's solubility guidelines. The inlet of municipal wastewater was provided by the Henriksdal wastewater treatment plant (Stockholm, Sweden). The inlet wastewater (InletWW) was taken directly from an entry tank with the following composition: COD 500–700 mg/L, $N_{tot}$ 40–50 mg/L, $P_{tot}$ 4.0–5.0 mg/L.

### 2.2. Isolation and Selection of Fungi from Municipal Wastewater

Potato dextrose (PD) agar (Oxoid, Cheshire, UK), in the presence of CAR, DIC, IBU, and SUL, was used for fungal isolation from the InletWW. The pharmaceutical compound with the final concentration 5 mg/L of selected chemical was added immediately after the synthetic wastewater media was poured into dishes. Briefly, the wastewater sample of 100 μL was spread on the plate and incubated for 7 days at 25 °C. After incubation, the isolates obtained were re-streaked on the new agar plate to isolate ten pure cultures for further experiments, and the colony size (%) of isolated fungi was determined in triplicate. Shortly, the fungal growth among strains was estimated qualitatively by taking images of the mycelia on the agar plate with a digital photo camera (Kodak, Rochester, NY, USA) and manually by measuring the colony size with a ruler (mm). The diameter of agar plate was accepted to be 100 % (90 mm). Thus, the colony size (%) was calculated using the formula: colony size (%) = (colony size (mm) × 100)/90 (mm).

For a t-test (two-tailed distribution; significance level ≤ 0.05) and data analysis, MS Excel 2013 was used.

### 2.3. Removal of Pharmaceutical Substances in Synthetic and Municipal Wastewater Media

The removal efficiency of DIC and CAR by fungal isolates was evaluated under sterile conditions, adjusting the pH of 5.5 and 6.3 (1 M HCl acid). Municipal wastewater (InletWW) was used as a negative control.

Firstly, a 250 mL glass flask was filled with 50 mL of a sterile synthetic wastewater medium (0.2 g $K_2HPO_4$, 0.8 g $KH_2PO_4$, 0.5 g $MgSO_4$, 0.2 g yeast extract in 1 L distilled water), pharmaceutical substance (the final concentration 5 mg/L of a selected chemical), and fungal inoculum (re-streaking on

a new agar plate, r = 20 mm). Finally, flasks were incubated in a shaking incubator (150 rpm) for a period of 10 days at 25 °C. The pharmaceuticals concentrations were periodically determined after incubation periods of 0, 3, 5, 8, and 10 days.

The removal efficiency of pharmaceutical substances from non-sterile municipal wastewater was conducted, using the most promising fungal isolate. Briefly, all experiments were performed under non-sterile conditions, as mentioned earlier. To achieve a higher initial concentration of fungal biomass, the K1 carrier units for a biofilm formation were used (one carrier unit per 1 mL) [16]. After growing biomass on carriers with a PD broth medium (Oxoid, United Kingdom) for 5 days, the fungal biomass was separated and added to non-sterile municipal wastewater. Finally, an additional investigation of the pharmaceutical removal efficiency under non-sterile conditions was performed for 48 or 72 h, and samples were taken at 0, 3, 6, 9, 12, 15, 18, 21, 24, 36, 48, and 72 h. All samples were filtered for further high-performance liquid chromatography (HPLC) and laccase activity analysis. *T. versicolor* was used as a positive control. All experiments were carried out in duplicate or triplicate.

### 2.4. Biosorption Test

Fungal biomass of the isolate was cultivated in the PD media (Oxoid, United Kingdom), incubating in a shaking incubator (50 rpm) for 5 days at 25 °C. Subsequently, half of the flasks were double autoclaved for 15 min at 121 °C to establish a heat-killed control (dead fungal cells). After autoclaving, the pharmaceutical substance was added to the live and dead fungal biomass of a concentration at 2.5 mg/L. Two additional negative controls—a control without chemical compounds and a control without fungi—were prepared to compare the experimental observations in order to be completely linked to the bioremoval induced by fungi. Finally, the samples for the HPLC and laccase activity analysis were taken at 0, 3, 6, 9, 12 and 24 h. The experiment was carried out in duplicate [17].

### 2.5. Analytical Procedure of Enzyme Activity and HPLC

The activity of laccase was measured spectrophotometrically, using the standardized procedure of the enzymatic assay for laccase by Sigma-Aldrich (Germany). Shortly, the test reaction consisted of 2.2 mL of a 100 mM potassium phosphate buffer ($KH_2PO_2$, pH 6.0), 0.5 mL of laccase from *T. versicolor* (crude powder, ≥50 units/mg solids, Sigma-Aldrich) and 0.3 mL of a 0.216 mM syringaldazine solution ($C_{18}H_{20}N_2O_6$, Sigma-Aldrich). After the reaction, absorbance changes were measured for 10 min at 530 nm. The measurements were carried out in triplicate.

The concentration of pharmaceuticals was measured by the HPLC system of an Alliance 2695 Separation Module (Waters, Milford, MA, USA) using 2996 Photodiode Array detector (Waters, Milford, MA, USA). Chromatographic separation was achieved with the Nova-Pak $C_{18}$ column (4 μm, 3.9 × 300; Water, USA) using a flow of 0.5 mL/min. The mobile phase was 70% *v/v* methanol and a 30% *v/v* 20 mM phosphate buffer (pH 2.5) (Sigma-Aldrich, Darmstadt, Germany). All results were analyzed, using Empower 3 Chromatography Data Software (Water, USA).

### 2.6. Identification of Fungal Isolates

Pure culture of an isolated fungal strain was sent for molecular-based identification to the Fungal Biodiversity center of the Westerdijk Fungal Biodiversity Institute (Utrecht, The Netherlands).

## 3. Results and Discussion

### 3.1. Isolation of Fungal Strains

From the inlet wastewater sample, ten fungal colonies were randomly selected and isolated to pure culture (Figure 1a). Isolated fungi were grown on PD agar in the presence of all selected pharmaceutical compounds, such as carbamazepine (CAR), diclofenac (DIC), ibuprofen (IBU), and sulfamethoxazole (SUL). Based on growth efficiency, three fungal isolates, F8, F9, and F10, were excluded from further analysis, due to the slow growth during the incubation period of 7 days (data not shown). Meanwhile,

the fungal strain F4 showed the highest growth on PD agar with/without pharmaceutical substances after 7 days of incubation (Figure 1a,b). Moreover, there was no statistically significant difference among fungal isolates F1, F3, F5, F6, and F7 ($p > 0.05$) when colony sizes of the fungi growth efficiency, on PD agar with or without pharmaceuticals, were compared. However, fungal isolates F2 and F4 demonstrated a relatively higher growth efficiency when the fungi were grown on PD agar without pharmaceuticals after 7 days of incubation.

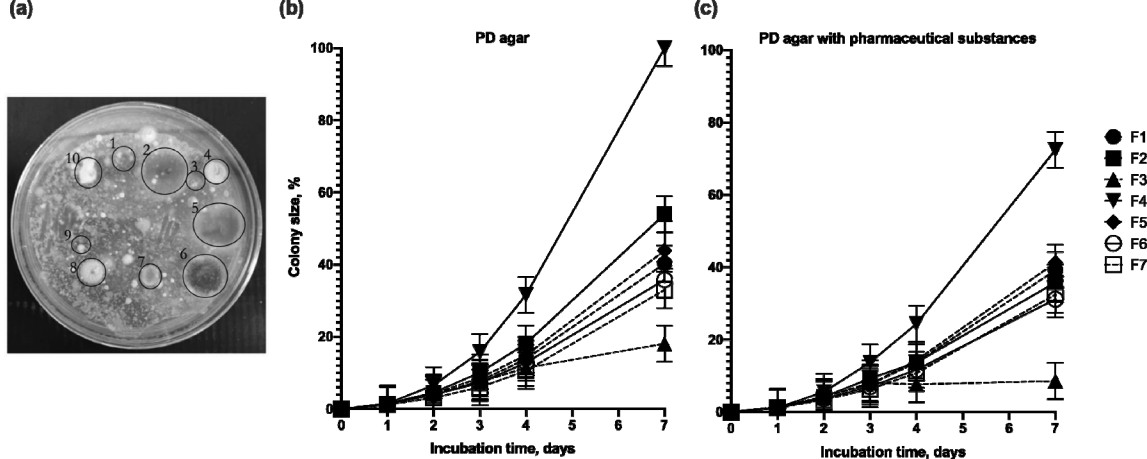

**Figure 1.** (**a**) Fungal isolates from municipal wastewater on a potato dextrose (PD) agar plate in the presence of carbamazepine (CAR), diclofenac (DIC), ibuprofen (IBU), and sulfamethoxazole (SUL); (**b**) Colony size (%) of isolated fungi on PD agar without pharmaceutical substances during the incubation period of 7 days; (**c**) The colony size (%) of isolated fungi on PD agar with pharmaceutical substances during the incubation period of 7 days.

In this study, the compounds selected cover a wide range of chemical structures. For instance, DIC has an active group of chlorine, while CAR has a group of amine. Meantime, SUL has a combination of active groups in their molecular structure. This variety of chemical structures might inhibit fungal growth in the presence of chemical compounds [8]. Therefore, this might explain why different colony sizes were observed in this study for each of the ten isolates.

Overall, a total of seven fungal isolates was used for further study to investigate their ability to remove pharmaceutical substances.

### 3.2. Removal of Pharmaceuticals by Fungal Isolates in Synthetic Wastewater Media

In this study, DIC and CAR were selected as model compounds, due to high consumption levels in various European countries and their appearance in wastewater treatment plant effluents [18]. The results in Figure 2 indicate the efficiency of removal by *T. versicolor* for DIC and CAR, comparing to fungal isolates F1, F2, F3, F4, F5, F6, and F7 in a synthetic wastewater media at a pH of 5.5. Results for fungal strains F4, F6, and F7 showed a relatively high removal efficiency (>80%) for CAR while *T. versicolor* removed <20% of this substance after 3 days of the incubation (Figure 2a). At the same time, fungal isolates F3 and F4 were able to remove >80% of DIC after 3 days of incubation while *T. versicolor* demonstrated complete removal (>99.9%) of DIC after 5 days of incubation (Figure 2b). Finally, fungal isolates F1, F2, and F5 showed a relatively low removal efficiency (<20%) of both pharmaceuticals throughout the incubation time. Therefore, the fungal isolates F3, F4, F6, and F7 were selected for a further investigation, e.g., to find out how the pH affects the removal efficiency of these strains.

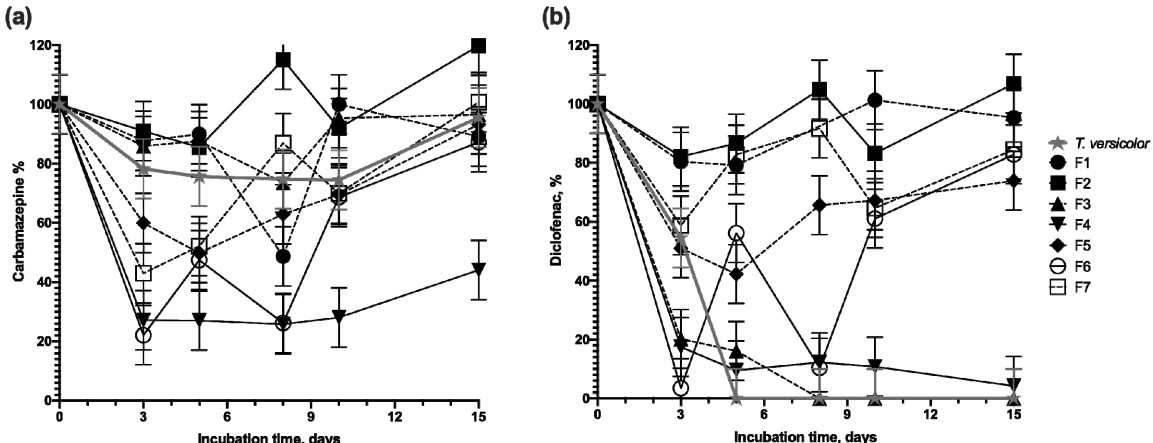

**Figure 2.** The removal efficiency (%) of CAR (**a**) and DIC (**b**) from a synthetic wastewater media of selected fungal isolates compared to *T. versicolor*.

### 3.3. pH Effect on Removal Efficiency

Previous studies have reported that the pH level has been determined to be one of the most important factors for enzyme production in fungi [19]. Additionally, the pH of 5.5 has been reported as the most relevant pH value for white-rot fungi growth and enzyme production, especially for *T. versicolor* [19] while the pH of 6.3 presents a natural pH value of synthetic wastewater media. Therefore, during this study, two pH values were selected, 5.5 and 6.3 respectively.

The results of Figure 3 showed the pH effect on CAR and DIC removal efficiency. The results of CAR indicated a relatively low (<20%) removal efficiency for both pH values of fungal strains F4, F6, and F7; there was no statistical difference ($p < 0.05$). However, a fungal isolate F3 and *T. versicolor* showed >25% removal of CAR at the pH 6.3, while no removal activity was obtained at the pH 5.5 after 3 days of incubation (Figure 3a,b). Overall, the results indicated difficulties to obtain relatively high removal efficiency (>80%) from CAR by tested isolates. Therefore, a further investigation is required to understand the removal mechanisms for this compound.

The results were derived to measure the DIC removal efficiency, showing that the fungal isolate F3 could completely (>99.9%) remove this pharmaceutical after 6 days of incubation at pH 5.5, while complete reduction of DIC at pH 6.3 was obtained after 10 days of incubation. Furthermore, the same result was observed for *T. versicolor* (Figure 3c,d). Overall, the fungal isolate of F3 has demonstrated the potential to remove DIC efficiently and successfully under sterile conditions. Thus, further in this work, the removal efficiency for DIC in non-sterile municipal wastewater with the isolate F3 was examined.

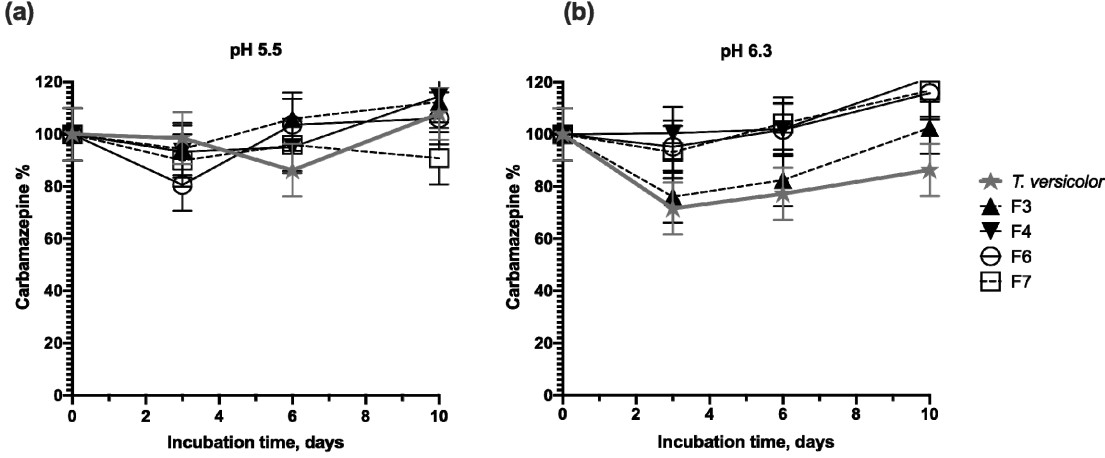

**Figure 3.** *Cont.*

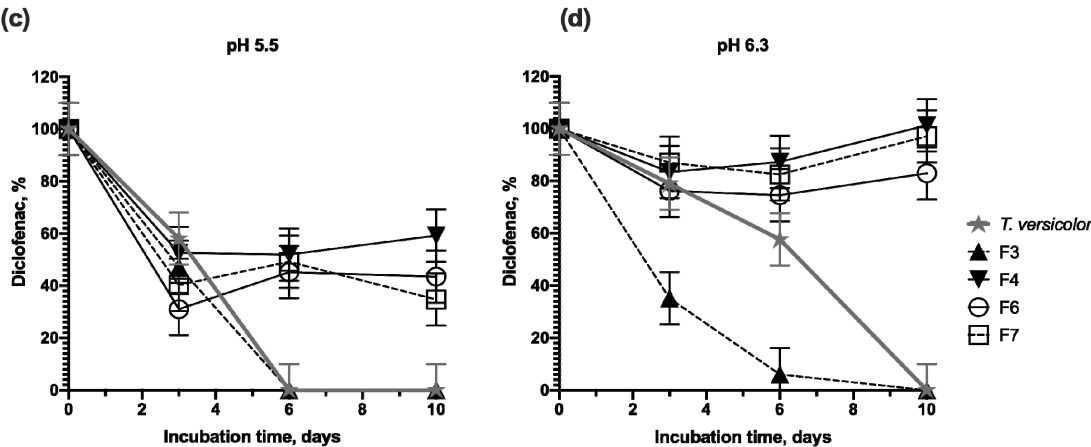

**Figure 3.** The removal efficiency (%) of CAR at (**a**) the pH 5.5 and (**b**) the pH 6.3 from synthetic wastewater media; The removal efficiency (%) of DIC at (**c**) the pH 5.5 and (**d**) the pH 6.3 from a synthetic wastewater media.

### 3.4. Removal of Pharmaceuticals by Fungal Isolates in Municipal Wastewater

Over the last decade, most studies on fungi have been conducted on an autoclaved media [14,20,21]. Therefore, the results were not always relevant to situations where fungi are applied to raw wastewater conditions [1]. Accordingly, biomass of the isolate F3 was transferred to non-sterile wastewater and the removal efficiency of DIC was analyzed. Based on previous results from this study, the pH of municipal wastewater was corrected from 7.8 to 5.5. However, the effect of a pH of 7.8 on the removal efficiency was also investigated to see the isolate potential for applications in wastewater treatment processes without a pH correction.

Figure 4 demonstrates the removal efficiency for DIC of the isolate F3 from non-sterile municipal wastewater, compared to *T. versicolor* and municipal wastewater (InletWW) as a negative control without a selected fungal strain.

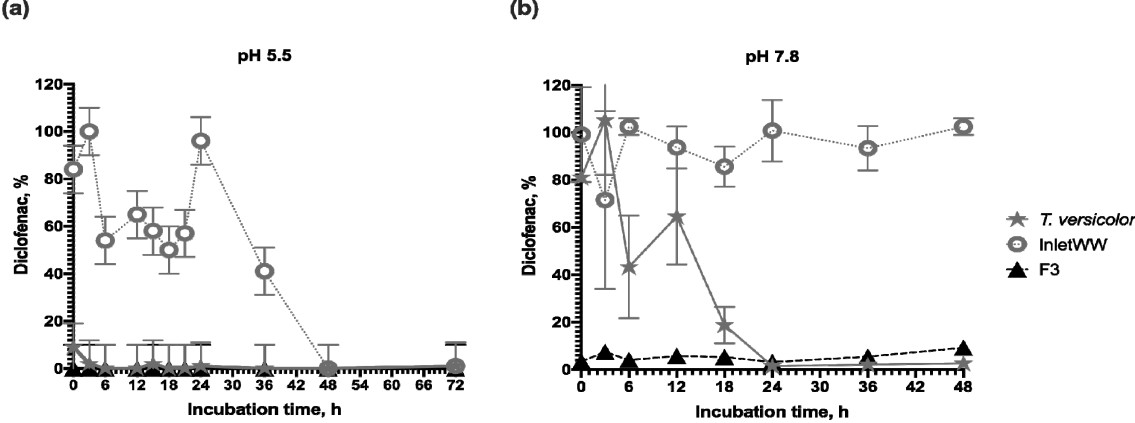

**Figure 4.** The removal efficiency (%) of DIC at (**a**) the pH 5.5 and (**b**) the pH 7.8 from non-sterile municipal wastewater.

When evaluating the DIC removal efficiency after fungal treatments at various pH values for a non-sterile wastewater sample, it can be stated that the isolate F3 can remove >95% of DIC for both pH values for the entire incubation time; there was no statistically significant difference between removal efficiency for both pH values ($p > 0.05$). In contrast, *T. versicolor* demonstrated a relatively slower DIC removal efficiency at the pH 7.8 after 24 h of the incubation time, compared to the pH 5.5 when DIC was completely removed (>99.9%) after 3 h of an incubation period. Furthermore, non-sterile

wastewater (InletWW) used as a negative control showed an ability to remove DIC at pH 5.5 after 48 h of the incubation time while no reduction was observed at pH 7.8. Furthermore, Figure 5a shows the pH changes during the incubation time with the isolate F3 where the pH was decreased from 7.8 to 3.5 for both samples immediately after the incubation started. This might be explained by the biological metabolism where the acetic acid and alkali might be produced to assimilate nutrients [22,23]. However, further investigation needs to be conducted to better understand the metabolism behavior of the isolate F3. At the same time, wastewater with *T. versicolor* demonstrated a reduction of the pH level from 7.8 to 5.5 after 24 h of the incubation period, when the total removal of DIC was also accomplished ( Figures 4a and 5a). Similar results have been observed in a previous study of *T. versicolor* where the maximal removal efficiency for DIC was observed at the pH 5 [24]. Meanwhile, wastewater without a fungal inoculum as a negative control did not show any pH changes throughout the incubation time (data not shown).

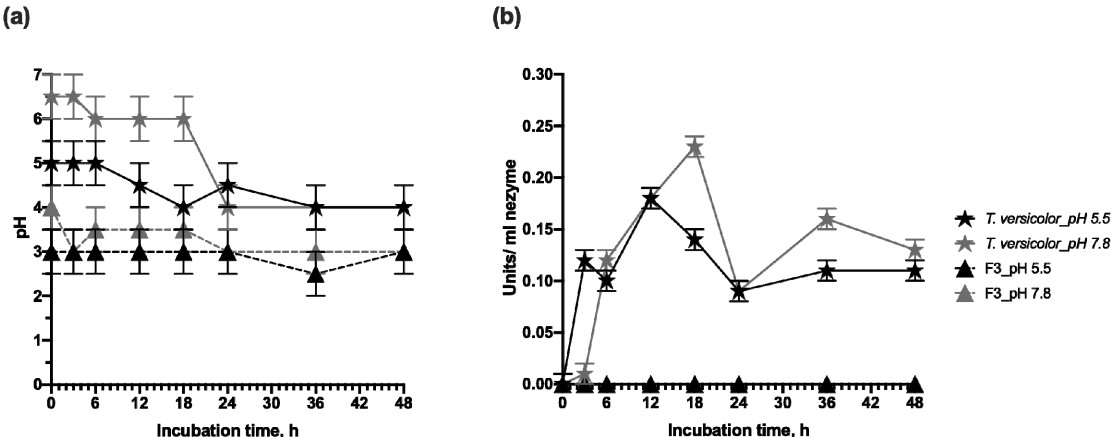

**Figure 5.** (**a**) pH changes in non-sterile municipal wastewater during the incubation period; (**b**) enzyme activity of laccase in non-sterile municipal wastewater during the incubation period.

Fungi have a variety of strategies to counteract pharmaceutical compounds, e.g., enzymatic processes such as biosorption as well as biotransformation and biodegradation mediated by enzymatic systems [25]. Therefore, in order to better understand the removal mechanisms for the isolate F3, the enzyme activity of laccase was measured to better understand the relationship between the enzyme activity and the removal efficiency for DIC. Laccase has been reported as an enzyme that can degrade a variety of pharmaceutical compounds, e.g., *T. versicolor* has shown an ability to remove ibuprofen and carbamazepine with the laccase enzyme [14].

The results indicated that there was no enzyme activity detected for the isolate F3 and wastewater without a fungal inoculum, while *T. versicolor* produced laccase for the entire incubation period for both non-sterile municipal wastewater samples with different pH values (Figure 5b). Furthermore, the live biomass of the isolate F3 did not present any laccase enzyme activity throughout the incubation period (Figure 6b). Moreover, the isolate F3 showed complete removal (>99.9%) of DIC for both live and dead fungal biomass, immediately after the biomass adjustment (Figure 6a), indicating that the removal of DIC could be due to the biosorption mechanism. In contrast, the results of the biosorption test indicated that *T. versicolor* used both strategies—biosorption and enzyme production—to remove DIC (Figure 6a,b). Finally, the results from negative controls with DIC did not show any DIC removal throughout the incubation period in the PD broth. Therefore, the results obtained could be induced by fungi (data not shown).

Finally, the isolate F3 was identified as *Aspergillus luchuensis*. Overall, these results have shown that *A. luchuensis* has a higher removal efficiency in a non-sterile wastewater sample without a pH correction than *T. versicolor*. Thus, *A. luchuensis* has a high potential for use in industrial wastewater treatment due to minimized specific pH requirements. Furthermore, to the best of our knowledge,

this is the first study where *A. luchuensis* has been reported as a promising strain for wastewater treatment in order to remove pharmaceutical substances. However, the requirements for nutrients and temperature need to be investigated; the impact of the treatment cost should be evaluated in further studies.

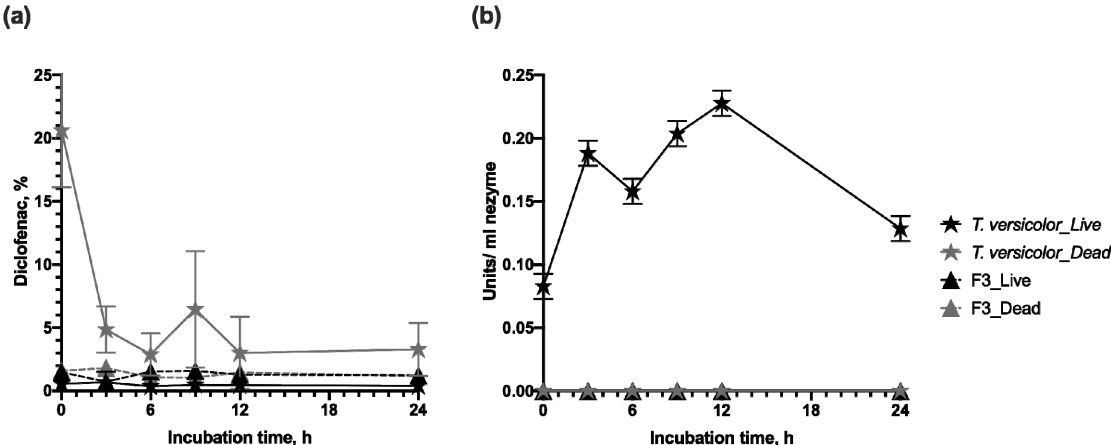

**Figure 6.** (**a**) The removal efficiency (%) of DIC from the biosorption test of live and dead fungal biomass by the isolate F3 and *T. versicolor*; (**b**) Enzyme activity of laccase from the biosorption test of live and dead fungal biomass by the isolate F3, compared to *T. versicolor*.

## 4. Conclusions

This study of fungi isolated from municipal wastewater demonstrated isolates' ability to grow in the presence of pharmaceuticals such as CAR, DIC, IBU, and SUL. A high removal efficiency rate was observed in the fungal isolate *Aspergillus luchuensis*, where complete (>99.9%) removal of DIC was observed in a synthetic wastewater media after 10 days of the incubation period, while >98% of DIC was removed in non-sterile wastewater without a pH correction. Isolates showed a lower removal efficiency of CAR compared to DIC. The removal mechanism for DIC of *A. luchuensis* is proposed to be a biosorption strategy. Overall, the results indicated *A. luchuensis* is a promising candidate to remove DIC from wastewater with a typical pH of 7.8, compared to *T. versicolor* which has shown a relatively higher removal efficiency at pH 5.5.

**Author Contributions:** T.J. and G.K.R. devised the project, the main conceptual idea, and proof outline. C.O. carried out the experiments. B.D. design the experiments and wrote the manuscript with support from C.O., T.J., and G.K.R. All authors have read and agreed to the published version of the manuscript.

**Funding:** The work was supported by the Waterchain project funded by the EU-Interreg Central Baltic Region.

**Acknowledgments:** Authors of the article would like to extend their sincere gratitude to the Riga Technical University for the funding provided to the doctoral student Brigita Dalecka.

**Conflicts of Interest:** The authors declare no conflict of interest.

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
