# Peer review of "Isolation of Fungal Strains from Municipal Wastewater for the Removal of Pharmaceutical Substances"

_water, doi:10.3390/w12020524_

Round 1

Reviewer 1 Report

I have reviewed the manuscript by Dalecka and colleagues. The authors have isolated 10 fungal strains from municipal wastewater and have tested the ability of selected strains to remove pharmaceutical substances (i.e. carbamazepine – CAR – and diclofenac – DIC) in pure culture conditions and in real wastewater. One isolate, identified as Aspergillus luchuensis, showed the highest removal abilities. The authors also suggested that the main DIC removal mechanism of A. luchuensis was by absorption. The manuscript is well written and well organized, furthermore the topic of the manuscript can be relevant for the journal. However, I think that the claims are not fully supported by the data since important controls are missing, and a better discussion can be useful.

General comments:

Removal of pharmaceuticals from synthetic medium and biosorption experiment. Abiotic controls, in addition to the heat-killed control, are needed to ensure that the removal of pharmaceuticals is not due to abiotic processes. Inoculation procedure for the synthetic medium test, for the municipal wastewater medium and for the biosorption test. In order to compare the removal efficiency of different strains the initial biomass should be the same. How can the authors ensure that with the used protocols the initial biomass is the same for different strains?

Specific comments:

Lines 45-46. Can the authors explain this statement? It is not clear why the strain tested in this study can reduce the cost. Line 67. Which is the concentration of pharmaceuticals in the culture medium? How were the standard solutions (5 mg/l, line 60) added to the medium? Line 70. How was the “colony size (%)” calculated? Line 74. Why were only CAR and DIC tested? Line 76. I expected that the synthetic wastewater was a liquid medium. Why was agar added? Lines 113-114. Please add more information. Furthermore, since the result for only one strain was reported please change “strains” to “strain” (line 113). Why is the growth different with pharmaceuticals? Please discuss. Lines 137-144. Were the differences significant? Figure 2. The figure is difficult to read. I suggest to find another way to report the data (e.g. one figure with CAR and DIC for each strain) Line 187. Please add this control in the Materials and Methods section. Lines 201-204 and Figure 5a. Why were the pH results from the negative control (i.e. wastewater without inoculum) not reported? Please add for a better comparison. Lines 205-206. Please, improve the discussion. Lines 206-207. The sentence seems strange. Please check. Lines 213-216 and Figure 5b. Why were the enzyme activity results from the negative control (i.e. wastewater without inoculum) not reported? Please add for a better comparison. Lines 218-222. Please, improve the discussion referring to the literature. Lines 231-232. The sentence seems strange. Please check. Lines 239-240. Please, use the abbreviations for the pharmaceuticals. Lines 243-244. The removal mechanism was proposed only for DIC. Please make it clear in the Conclusions.

Author Response

Dear Reviewer,

Best regards,

Brigita Dalecka

Reviewer 2 Report

In this study Dalecka et al isolate different fungal strains from a wastewater sample and test their ability to degrade three different pharmaceutical compounds. This is a basic study that might be publishable, but the authors need to make many improvements.

The language mostly okay but I suggest a native speaker review as there is many grammar errors and it requires me being very focused to read through. I also spotted a few typos e.g. statically = statistically.

Abstract and introduction: Fungi are not a wastewater treatment method by themselves, they occur within a complex mixture of microorganisms that metabolizes biodegradable constituents in wastewater, particularly during secondary wastewater but also already in the sewer system. The role of fungi and their importance in wastewater treatment should be better described e.g. what their exact role is within the community. Then, their role in degrading pharmaceuticals (and other hard to degrade compounds in wastewater) should be explained, maybe in literature there is some figures on how important their contribution is, for example during activated sludge treatment. Perhaps there is not much research but look it up.  Fungi have generally a different enzymatic toolset than bacteria that allows them to cleave certain bonds more efficiently. I’m not up to date with the literature in the field and the current state of research but authors should add more information on this to better set the background. Also, discuss potential limitations of fungi (slower growth, competition, oxygen requirement and pH). Building introductory story line around this and using more recent and more specific research articles (look also for reviews in the area) would greatly improve readability and value of this work, while the actual results are modestly exciting.  

Mention that fungal strains were isolated first, before explaining you tested them. Abstract for small studies best to follow sequence of study steps as presented in manuscript.

‘We found that A. luchuensis is able to lower the pH level in a non-sterile municipal wastewater sample from 7.8 to 3.5.’ Isn’t that a trivial finding, worth mentioning in an abstract?

‘Therefore, we assume the isolate is a promising candidate to remove pharmaceutical compounds in a wastewater treatment process minimizing the specific requirement – the pH’ – I do not understand this conclusion, why is pH a specific requirement? Can you write a better concluding sentence e.g. including rather the pharmaceutical removal?

Material and Methods
Section 2.2. Is this a typical method to isolate Fungi from liquid media? Perhaps add reference.
Section 2.3 and results section. I am unclear about what type of removal mechanisms we are looking at is it metabolism (so pharmaceuticals as sole carbon/energy source) or co-metabolism, the fungi have a different energy source and the pharmaceuticals are degraded alongside? Shape out better.   

Compare research section with final paragraph of introduction that provides summary of study, some aspects of study such as identification of isolates, biosorption and enzyme activity measurements might be useful to mention here, also better embed in introduction, currently to much focused on pH.

Section 2.6. Why not listing identity of al 10 isolates in results section? Data not good, not available?

Results section: overall more interpretation of results is needed. Currently very focused on presenting results, e.g. line 158 – 164. Why CAR removal is low, how does this compare to the literature. Perhaps could also argue with molecular structure of substance and accessibility by enzymes, typical degradation mechanisms by fungi? Just ideas….

Why role of laccase was investigated has to be better explained.

Conclusion: Adsorption strategy: Careful with this statement. Why would a Fungi have an adsorption strategy for a pharmaceutical, needs further explanation in main text with literature reference.

Final sentence of conclusions as well careful. Rather write that strains such as the one found might be responsible for degradation of pharmaceutical as they are most active in typical pH range of wastewater and occur there, and therefore better than other strains that are more active at other pH. It would be great if the pH relationship of Fungi was also properly reviewed an introduced in the introduction.

Author Response

(The authors gave the same response as above.)

Reviewer 3 Report

The paper is interesting as the authors used two types of fungi for the removal of pharmaceutical compounds from wastewater. I would like to give a chance for authors to improve the paper. I suggest major revision.

Line 12: The fungi is a method?

Do not write "we" in an academic paper.

The methodology should be cited.

It is better to mention all the effecting parameters which have an influence on the removal. The authors only mentioned about pH. 

The main problem is the lack of enough experiments. The effect of temperature also should be added.

The discussion is weak and there is no comparison with the published studies in this field.

Did the authors use both of the fungi in the same experiment? It is good to see what are the effect of them on the removal.

The references should be expanded to more studies (at least 25).

Author Response

(The authors gave the same response as above.)

Round 2

Reviewer 1 Report

The authors have addressed my comments and improved the manuscript. I appreciate the work done by authors and I have only few of minor comments.

I thank the authors for mentioning that negative controls were performed (lines 107-109). I am just missing if abiotic loss of pharmaceuticals was observed or not in the abiotic experiment without heat-killed cells. It might be useful to include the volume of the stock solutions of pharmaceuticals added (line 70-71). Maybe it was not clear from my comments to the previous version. I think that some more info on the methods used for the identification of the fungal strain (lines 125-127) can be useful. Just a couple of lines or a reference to understand what was done.

Author Response

Dear reviewer,

The missing information has been added to the text and the revision in the manuscript has been highlighted in red colour (Line 70- 71, 126-127).

During this study, the removal of the pharmaceutical compound without heat-killed fungal biomass (only in PD media) was not observed.

The text has been double-checked with a native speaker. The revision in the manuscript has been highlighted in red colour.

Best regards,

Brigita Dalecka

Reviewer 2 Report

The authors addressed my comments, therefore I am happy with this version. Though the language could be still improved. 

Author Response

Dear reviewer,

The text has been double-checked with a native speaker. The revision in the manuscript has been highlighted in red colour.

Best regards,

Brigita Dalecka

Reviewer 3 Report

The paper needs the following changes:

The introduction is too short and needs more critically writing about the gap of research. Novelty should be more clear. The discussion is too weak. References should be modified and updated.

Author Response

Dear reviewer,

The text has been double-checked with a native speaker. The references have been formatted according to journal guidelines. The revision in the manuscript has been highlighted in red colour.

Best regards,

Brigita Dalecka